# SHADE: Spectral Hallucination Detection via Dual Spectral Decompositions

## Abstract

Large language models often produce confident but unsupported statements. Detecting such hallucinations from internal signals is essential for reliable systems. Existing attention-based methods either summarize weights with local statistics or adopt Laplacians (e.g., $D_{\text{out}} - A$) whose guarantees and applicability break outside strictly causal, square attention. We seek a single, rigorous framework that scales across architectures and attention types while yielding interpretable indicators of grounding. We introduce SHADE, a unified spectral approach that models attention with two standard operators: (i) a random-walk operator $L_{\text{rw}} = I - A$ that quantifies diffusion/leakiness, and (ii) a degree-normalized cross-operator $M = D_Q^{-1/2} A D_K^{-1/2}$ that quantifies query–key coupling. The resulting features are mathematically rigorous and physically interpretable, with clear operator semantics that map to failure modes associated with hallucination: *diffusion* (from $L_{\text{rw}}$) quantifies probability leakiness; *conductance* (via a symmetric/PSD Laplacian, e.g., Chung's) captures expansion/connectedness; and *energy/alignment strength* (from the SVD of $M$) quantify total coupling and the dominance or fragmentation of coupling modes. The formulation applies unchanged to encoder, decoder, and encoder–decoder settings, including rectangular cross-attention and masked sub-blocks. Evaluated on GPT-2, FLAN-T5, and Phi-2 across HaluEval and TruthfulQA, SHADE consistently surpasses token-probability and LapEigvals-style baselines, delivering strong discrimination and calibration alongside interpretable spectra. By grounding hallucination detection in standard spectral operators with physically meaningful interpretations, SHADE offers an explanatory basis for *where* and *how* hallucinations originate. This suggests two practical avenues: (i) training-time regularization to suppress emergent hallucinatory patterns, and (ii) a deployable *hallucination risk score* with mode-level rationales (by layer/head and prompt span) that end users and developers can act on.

## 1 Introduction

Large language models (LLMs) are increasingly used in settings where errors carry real-world cost, yet they can generate confident but unsupported content. Practitioners need *actionable* signals that (i) indicate when a response is at risk of hallucination, (ii) explain *where* the problem originates (within the model's attention dynamics), and (iii) suggest *how* to intervene at training or deployment time.

We advocate a spectral view of attention that yields compact, interpretable indicators of grounding. Rather than treating attention weights as opaque heatmaps, we summarize attention as standard operators with clear semantics—diffusion, expansion, and coupling energy—to produce an *attention health* readout per layer/head and per prompt span. The goal is a deployable *hallucination risk score* accompanied by short, mode-level rationales.

We instantiate this vision in SHADE, a unified spectral framework built on two standard operators: (i) a random-walk view that quantifies how attention diffuses away from tokens (*leakiness*); and (ii) a degree-normalized query–key coupling view that measures the strength and dimensionality of alignment between what the model is trying to produce and the

evidence it conditions on. Both operators are computed in a batched, head-aware manner and the latter applies unchanged to encoder, decoder, and encoder–decoder architectures, including rectangular cross-attention and masked sub-blocks.

Across multiple model families (GPT-2, FLAN-T5, Phi-2) and benchmarks (HaluEval, TruthfulQA), SHADE delivers strong discrimination and calibration relative to weight-only or Laplacian baselines while providing interpretable spectra that localize failure modes. This enables two practical pathways: (i) training-time regularization that targets diagnosed modes (e.g., suppressing over-concentrated coupling), and (ii) deployment-time risk scoring with concise, layer/head-level rationales that users and developers can act upon.

**Illustrative scenarios**    We highlight three common attention pathologies and how SHADE surfaces them.

- **Prompt detachment during generation.** As decoding proceeds, the model gradually shifts mass to recently produced tokens and recycles its own content. *SHADE readout:* the coupling view reports a decline in leading-mode strength and early energy (weak evidence alignment), while the diffusion view shows persistently low leakiness away from recent tokens. *Action:* raise a risk alert; at training time, regularize to maintain coupling diversity on long generations.

- **Anchor collapse to BOS or a single context shard.** Many queries align to one key direction (e.g., BOS), creating a rank-1 pattern that "sounds fluent" but ignores context diversity. *SHADE readout:* high dominance of the top coupling mode and a sharp spectral gap (low effective/stable rank). *Action:* warn about brittle anchoring; during finetuning, penalize excessive mode dominance or encourage multi-mode coupling.

- **Context bottlenecks across topical regions.** The prompt contains distinct regions (e.g., two sources). Answers drift within one region and fail to integrate evidence across them. *SHADE readout:* the expansion view (evaluated via a symmetric/PSD Laplacian variant) exposes a small second eigenvalue consistent with a connectivity bottleneck across regions. *Action:* prompt- or retrieve-time intervention (e.g., re-ordering or segment bridging); training-time objectives that improve cross-region mixing.

**Key Contributions.**

- **Unified operator view.** A single spectral framework for attention that uses $L_{\mathrm{rw}}$ (diffusion/leakiness) and $M$ (degree-normalized coupling), applicable to self- and cross-attention, including rectangular blocks.

- **Rigorous, interpretable features.** Physically meaningful indicators—diffusion, conductance (via a PSD variant), and energy/alignment strength—that map naturally onto hallucination-related failure modes.

- **Architecture-agnostic and batched.** Head-aware, vectorized extraction that plugs into encoder, decoder, and encoder–decoder models without modification.

- **Empirical effectiveness.** Consistent gains over token-probability and LapEigvals-style baselines on HaluEval and TruthfulQA across three model families, with strong discrimination and calibration.

**Paper Organization.**    Section 2 situates SHADE among attention-based, spectral, and other related hallucination detection approaches. Section 3 details the operator foundations (random-walk and degree-normalized cross-operator) and the resulting feature semantics. Section 4 reports on the architecture-aware attention processing and experiments. We conclude with implications for regularization and deployable hallucination risk scoring.

## 2  RELATED WORK

**Scope and terminology.**    Hallucination detection methods broadly fall into three families: (i) analyses of *attention structure*, (ii) probes of *internal activations*, and (iii) *output-space* uncertainty measures. We use "hallucination" in the practical sense of producing unsupported

or incorrect content in context (Huang et al., 2025), acknowledging that perfect factuality is unattainable in general (Xu et al., 2024).

**Attention-structure analyses.** LapEigvals (Binkowski et al., 2025) introduces a future out-degree Laplacian $L_{\text{out}} = D_{\text{out}} - A$ (edge convention $A_{ij}: j \rightarrow i$) so that, under strict causality, eigenvalues are read off the diagonal. This convenience sacrifices spectral content (off-diagonal couplings/eigenvectors), lacks PSD/Hermitian guarantees, depends on absolute position/length, and does not directly cover bidirectional encoders or rectangular cross-attention. In contrast, we adopt *standard* operators with clear semantics and broad applicability: the random-walk operator $L_{\text{rw}} = I - A$ (Markov diffusion/leakiness) and the degree-normalized biadjacency $M = D_Q^{-1/2} A D_K^{-1/2}$ whose SVD quantifies query–key coupling. Our features are architecture-agnostic, batched, and apply unchanged to masked sub-blocks and rectangular attention.

**Origin of the degree-normalized cross-operator.** The operator $M$ is classical in several literatures: spectral co-clustering on bipartite graphs uses $D_r^{-1/2} A D_c^{-1/2}$ with SVD for partitioning (Dhillon, 2001; Zha et al., 2001), and correspondence analysis employs the same degree-weighting (with centering) followed by SVD. These precedents justify calling $M$ a *standard* degree-normalized biadjacency.

**Attention statistics and topological summaries.** Head-wise statistics (mean attention, entropy) aggregated by sequence models (Ogasa & Arase, 2025) offer simplicity but treat weights as absolute, ignoring how value geometry modulates effective contribution. Topological analyses compute Betti numbers on attention graphs (Bazarova et al., 2025); these capture global structure but are less directly tied to operator semantics. By contrast, our spectral operators yield physically interpretable readouts (diffusion, expansion via a PSD variant, and coupling energy/alignment strength) that we connect to failure patterns.

**Activation-based detectors.** Linear probes on "observer" activations can be effective but require auxiliary models and labeled data (O'Neill et al., 2025); unsupervised constraints can identify "truth directions" (Burns et al., 2023) but may oversimplify. Layer-wise drift and probing studies corroborate that internal states carry hallucination signals (Zhang et al., 2025). Our approach reads the *operators* that shape those states, providing earlier, interpretable signals without extra models.

**Output-space uncertainty and external validation.** Token-probability and semantic-uncertainty methods assess entropy or consistency over outputs (Quevedo et al., 2025; Farquhar et al., 2024); they capture the full effect of model geometry but often require careful calibration and, for semantic methods, multiple inference passes. External-validation approaches (KGR, RAG with hallucination-aware tuning, self-correction) ground or verify outputs using databases, retrieval, or tools (Guan et al.; Song et al., 2024; Asai et al., 2024; Dhuliawala et al., 2024; Gou et al., 2024; Chern et al., 2023). These are complementary to our goal: provide *internal*, operator-based signals that are lightweight, interpretable, and architecture-agnostic.

*Takeaway.* Prior work either compresses attention into local statistics, relies on Laplacians with limited guarantees, or moves outside the model. We instead leverage two spectral operators—$L_{\text{rw}}$ and the degree-normalized biadjacency $M$—to obtain rigorous, interpretable indicators that scale across attention types and transformer architectures.

## 3  DUAL SPECTRAL ANALYSIS

### 3.1  THEORETICAL HYPOTHESIS: EQUILIBRIUM STRUCTURAL INFORMATION

Our hypothesis is that hallucination detection benefits from equilibrium structural information in attention patterns. Unlike approaches that analyze post-hoc activation patterns or output distributions, we hypothesize that the structural properties of attention matrices at equilibrium, expressing how information flows and couples within the attention mechanism

itself, contain signatures of failure modes that are both mathematically rigorous and physically interpretable.

Our central hypothesis is that attention patterns during hallucination exhibit distinct structural anomalies that manifest differently across dual mathematical perspectives. We theorize that analyzing attention through different spectral lenses will reveal:

1. **Flow anomalies**: Disrupted information propagation patterns detectable via directed graph spectral analysis
2. **Coupling anomalies**: Pathological query-key alignment patterns revealed through bipartite graph decomposition

### 3.2  Dual Spectral Perspectives

We hypothesize that hallucination detection requires understanding both *how* tokens influence future predictions (via attention received) and *which* tokens couple strongly (query-key matching). We hypothesize these represent complementary mathematical objects that capture complementary structural information about attention failures.

Let $A \in \mathbb{R}^{n_q \times n_k}$ denote the attention probability matrix from a single head after softmax, where rows sum to 1 on their allowed support. For self-attention (encoder-only and decoder-only models), $n_q = n_k = n$, while for cross-attention they may differ.

#### 3.2.1  Flow Analysis via Directed Graphs

Spectral limitations of LapEigvals. With the causal self-attention convention where $A_{ij}$ denotes attention received by $i$ from $j$, LapEigvals sets $L = D_{\text{out}} - A$ with $(D_{\text{out}})_{ii} = \frac{1}{n-i} \sum_{j>i} A_{ji}$. This choice ensures $L$ is lower-triangular, so its eigenvalues are the diagonal entries $(D_{\text{out}})_{ii} - A_{ii}$. While computationally convenient, this collapses the spectrum to two scalars per index (self-loop $A_{ii}$ and future-mass average), discarding all off-diagonal coupling. The operator is not a graph Laplacian in the usual sense (row/column sums are not zero), lacks PSD/Hermitian structure and a variational characterization, and its eigenvalues can be negative (e.g., the last token). Moreover, because $D_{\text{out}}$ averages only over future nodes and divides by $n - i$, the spectrum is inherently order- and length-dependent, introducing boundary artifacts. In contrast, standard constructions such as the random-walk Laplacian $I - A$ or Chung's directed Laplacian preserve Markov or expansion semantics, yield real spectra with established guarantees, and retain sensitivity to global coupling patterns beyond the diagonal.

$L_{\text{rw}}$ is spectrally better with a minimal change. Replacing LapEigvals' $L = D_{\text{out}} - A$ by the normalized random-walk Laplacian $L_{\text{rw}} = I - A$ (the normalization cancels due to the row stochasticity of $A$) yields immediate spectral guarantees. Because $A$ is row-stochastic, $\text{spec}(A) \subset \{z : |z| \le 1\}$, so $\text{spec}(L_{\text{rw}}) = \{1 - \mu\}$ lies in the disk $|z - 1| \le 1$, with $\text{Re}(z) \in [0, 2]$; in the causal triangular case one even has $\lambda_i = 1 - A_{ii} \in [0, 1]$. Moreover, $L_{\text{rw}}\mathbf{1} = 0$, providing a canonical zero eigenvalue and the usual spectral-gap semantics of Markov operators. Thus, with a single, simple change, one obtains bounded spectra, standard random-walk interpretations, and permutation-stable behavior, while retaining the causal "read-off-the-diagonal" convenience.

On top of this operator we extract vectorized, per-head/per-layer features: the ordered eigenvalues $\lambda_{1:k}$ (interpretable as self-leakiness $1 - A_{ii}$ in causal decoders), early spectral gaps $\Delta_i = \lambda_{i+1} - \lambda_i$, a trace $\sum_i \lambda_i = n - \text{tr}(A)$ (total leakiness budget), spectral radius $\max_i \lambda_i$, and an entropy family (von Neumann/spectral entropy, effective dimension, participation ratio, and flatness) computed on the non-negative spectrum. These statistics constitute theoretically motivated proxies for hallucination risk. In principle, prompt detachment and self/recency fixation manifest as persistently small $\lambda_i$, low trace, and low spectral entropy/participation ratio (concentrated spectrum with weak expansion), whereas well-grounded behavior exhibits moderate $\lambda_i$, larger gaps, and higher entropy (diverse reliance on context). For non-causal square attentions (e.g., encoders), where $A$ is generally non-normal and eigenvalues may be complex, one can (i) keep $L_{\text{rw}}$ and base dispersion features on $|1 - \mu_j(A)|$ or on singular values, or (ii) switch to a PSD surrogate, e.g., Chung's directed

Laplacian or a symmetrized form after Sinkhorn balancing, to obtain real spectra and Cheeger-style interpretability; the same feature suite then applies unchanged. These claims are theoretical and call for empirical validation, but they provide a mathematically consistent upgrade over LapEigvals with minimal implementation burden.

### 3.2.2 Degree-normalized Biadjacency

To complement one-sided flow with a *two-sided* view that is agnostic to attention type and shape, we model (queries $\leftrightarrow$ keys) as a weighted bipartite graph with biadjacency $A \in \mathbb{R}^{n_q \times n_k}$. Let $d_i^Q = \sum_j A_{ij}$ and $d_j^K = \sum_i A_{ij}$. Define the *degree-normalized cross-operator*

$$M = D_Q^{-1/2} A D_K^{-1/2} \in \mathbb{R}^{n_q \times n_k}, \qquad (D^{-1/2})_{ii} = \begin{cases} d_{ii}^{-1/2}, & d_{ii} > 0, \\ 0, & \text{otherwise.} \end{cases} \tag{1}$$

For numerical stability one may use

$$\left(D_\varepsilon^{-1/2}\right)_{ii} = (d_{ii} + \varepsilon)^{-1/2} \quad \text{with small } \varepsilon > 0. \tag{2}$$

The SVD $M = U\Sigma V^\top$ exposes *two-sided coupling*: $U$ and $V$ are query/key modes; the singular values $\sigma_r$ measure coupling strength *after* degree effects are removed. This single framework covers self-attention ($n_q = n_k$) and cross-attention ($n_q \neq n_k$) without changing the mathematics.

**Bridge to spectral graph theory.** Placing $M$ in a graph-spectral wrapper requires no new assumptions: the symmetric block operator

$$\mathcal{B} = \begin{pmatrix} 0 & M \\ M^\top & 0 \end{pmatrix} \tag{3}$$

is a normalized bipartite adjacency whose eigenvalues are the signed singular values $\{\pm\sigma_r(M)\}$. Equivalently, the bipartite Laplacian $\mathcal{L}_{\text{bip}} = I - \mathcal{B}$ has spectrum $\{1 \pm \sigma_r(M)\}$ (plus copies of 1 when $M$ is rank-deficient). Thus, eigen-analysis of $\mathcal{L}_{\text{bip}}$ is informationally equivalent to the SVD of $M$ via the map $\lambda = 1 \pm \sigma$ and does not, by itself, create additional "complementarity." The formal statement and proof are given in Appendix C, Thm. C.1. *Proof sketch:* if $Mv_i = \sigma_i u_i$ and $M^\top u_i = \sigma_i v_i$, then

$$\mathcal{B} \begin{bmatrix} u_i \\ \pm v_i \end{bmatrix} = \pm\sigma_i \begin{bmatrix} u_i \\ \pm v_i \end{bmatrix}. \tag{4}$$

**Practical Implications.** In practice, we compute degree-robust coupling features from $\Sigma$ and $(U, V)$ (top-$k$ $\sigma_r$, energy concentration / stable rank, two-sided leverage scores with entropy/GINI, and cross-head/layer subspace overlap), batched across heads/layers. These are complementary to $L_{\text{rw}}$ eigenvalue features: $L_{\text{rw}}$ characterizes *one-sided diffusion/leakiness* of attention, whereas $M$ characterizes *two-sided alignment* between queries and keys. Together they yield attention-type-agnostic diagnostics; our hypothesis is that prompt detachment and brittle anchoring correspond, respectively, to low leakiness entropy/trace in $L_{\text{rw}}$ and to low $\sigma_1$ (or peaky key-side leverage) in $M$. Empirical validation is left to experiments.

**Degree-normalized coupling features.** Define $\|M\|_* = \sum_i \sigma_i$, $\|M\|_2 = \sigma_1$, $\|M\|_F = (\sum_i \sigma_i^2)^{1/2}$, and normalized energies $p_i = \sigma_i^2 / \|M\|_F^2$. We compute the following vectorizable features per head/layer:

- **Spectrum:** top-$k$ singular values $\sigma_{1:k}$; nuclear norm $\|M\|_*$ (total coupling); spectral norm $\|M\|_2$ (largest mode); Frobenius norm $\|M\|_F$ (overall energy); spectral gap $\gamma_1 = \sigma_1 - \sigma_2$; stable rank $r_{\text{stable}} = \|M\|_F^2 / \sigma_1^2$; decay ratios $\rho_i = \sigma_{i+1}/\sigma_i$.
- **Entropy/dispersion:** effective rank $r_{\text{eff}} = \exp(H)$ with $H = -\sum_i p_i \log p_i$; participation ratio $\text{PR} = 1/\sum_i p_i^2$; Gini on $\{p_i\}$; top-$m$ concentration $C_m = \sum_{i \leq m} p_i$ (e.g., $m \in \{5, 10\}$).
- **Condition/structure:** condition number $\kappa = \sigma_{\max}/\sigma_{\min^+}$; numerical rank $r_\tau = \#\{i : \sigma_i \geq \tau\}$; rank deficiency $r - r_\tau$; energy curves $E_k = \sum_{i \leq k} \sigma_i^2 / \|M\|_F^2$ (e.g., $k \in \{1, 3, 5, 10\}$).

*Rationale to hallucination.* These features summarize *degree-corrected* query↔key coupling. Low $\sigma_1$ / small $E_k$ (weak total/early energy) indicate **prompt detachment**; large $\sigma_1$ with large gap $\gamma_1$ but low $r_{\text{eff}}$ / PR and high $C_m$ / Gini indicate **brittle anchoring** to a narrow evidence subspace. Stable-rank growth and higher entropy reflect diversified grounding. We compute the same features for decoder cross-attention, encoder self-attention, and response→prompt blocks. (As with all statistical features, they provide principled proxies and require empirical validation.)

**Proposition 3.1** (Complementarity of flow and coupling perspectives)**.** *Let $A \in \mathbb{R}^{n \times n}$ be a nonnegative attention matrix with row sums 1 (softmax). Define the* flow *operator $L_{\text{rw}} = I - A$ and the* coupling *operator $M = D_Q^{-1/2} A D_K^{-1/2}$ (Eq. 1). Then the eigenvalue features of $L_{\text{rw}}$ (random-walk flow) and the singular-value features of $M$ (degree-normalized bipartite coupling) are, in general,* complementary*: they are invariants of different operators, respond to different equivalence classes (row-wise diffusion vs. two-sided alignment), and neither spectrum determines the other except in special cases (e.g., symmetric, doubly-stochastic $A$, where $M = A$ and $L_{\text{rw}} = I - A$ imply $|1 - \lambda_i(L_{\text{rw}})| = \sigma_i(M)$).*

*Proof sketch.* $L_{\text{rw}}$ is a (directed) Markov generator whose spectrum is tied to row-wise diffusion/leakiness, invariant under row renormalizations that preserve the Markov operator. In contrast, $M$ is a degree-normalized *biadjacency* whose SVD is invariant to positive diagonal scalings of rows and columns (absorbed by $D_Q, D_K$) and captures two-sided alignment. There is no algebraic map from $\text{spec}(L_{\text{rw}})$ to $\text{svd}(M)$ in general (different similarity classes, different invariances). The coincidence only arises under restrictive symmetries (e.g., $A = A^\top$ and doubly stochastic), where $M = A$ and the Hermitian dilation yields $\lambda(\mathcal{L}_{\text{bip}}) = 1 \pm \sigma(M)$. □

**Example (Uniform BOS attention).** Suppose all non-BOS tokens attend entirely to the BOS token 1 and BOS self-attends: $A_{11} = 1$, $A_{i1} = 1$ for $i > 1$, and $A_{ij} = 0$ otherwise (causal, row-stochastic).

- **Flow ($L_{\text{rw}}$):** $\lambda_i(L_{\text{rw}}) = 1 - A_{ii}$ gives $\lambda_1 = 0$ and $\lambda_i = 1$ for $i > 1$ (maximal leakiness away from self for $i > 1$), but it does not distinguish *where* the mass goes (all of it collapses onto BOS).
- **Coupling ($M$):** Column degrees satisfy $d_1^K = n$, $d_j^K = 0$ ($j > 1$), so $M$ has a single nonzero column with entries $1/\sqrt{n}$. Hence $\sigma_1(M) = 1$ and $\sigma_{r>1}(M) = 0$ — a clear rank-1 collapse (all queries aligned to one key mode).

Taken together, the features separate phenomena: $L_{\text{rw}}$ signals extreme self-leakiness (away from self-loops), while $M$ certifies loss of diversity in evidence coupling. This *complementarity* underpins our joint use of flow and coupling features for diagnosing prompt detachment vs. brittle anchoring.

## 4   Experiments and Results

Building on our theoretical framework, we develop architecture-specific applications of our dual spectral perspectives. Our detection pipeline is fairly generic and can accomodate different concrete implementations of the processing steps shown in figure 1.

### 4.1   Experimental Setup

**Feature computation.**   For each input sequence of length $n$, we record per-layer, per-head attention weights during inference. The attention processor is able to deliver any type of attention matrix downstream for further analysis. To ensure adequate spectral resolution we retain sequences with $n \geq n_{\min}$ (we set empirically $n_{\min}$ to retain 90% of the data). We construct two operator families. (i) *Laplacian path:* the (symmetric) normalized random-walk Laplacian $L^{(l)} = I - D^{-1/2} \bar{A}^{(l)} D^{-1/2}$ with $D = \text{diag}(\bar{A}^{(l)} \mathbf{1})$; in standard self-attention $D \approx I$ so $L^{(l)} \approx I - \bar{A}^{(l)}$. We extract eigenvalue-based features: top-$k$ eigenvalues (where $k = n_{\min}$), spectral gaps, low-order spectral moments, and von Neumann–type entropies. (ii) *SVD path:* the degree-normalized cross-operator $M^{(l)} = D_Q^{-1/2} \bar{A}^{(l)} D_K^{-1/2}$ for (queries, keys),

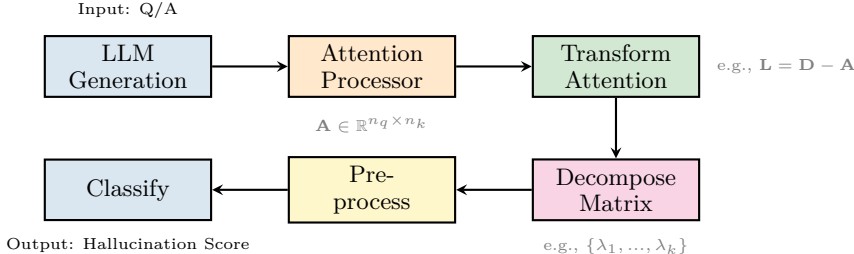

Figure 1

applied to self-attention blocks and, without modification, to rectangular cross-attention; we compute top-$k$ singular values, gaps, and summary functionals (e.g., Frobenius/nuclear norms, effective/stable rank).

**Preprocessing, classifier, and calibration.** We apply a fixed four-step preprocessing to all feature sets: (1) remove near-constants (variance $< \tau_{\text{const}} = 10^{-5}$); (2) prune collinearities via Pearson screening ($|\rho| > 0.95$); (3) standardize to zero mean/unit variance with $\epsilon = 10^{-8}$ for numerical stability; and (4) run adaptive PCA, retaining the smallest number of components that explain $\geq 85\%$ of variance, with at least 10 and at most 512 components ($d_{\text{final}} = \min(\max(10, d_{95\%}), 512)$). The resulting vector $\mathbf{x} \in \mathbb{R}^{d_{\text{final}}}$ is fed to a logistic regression detector $\Pr(y{=}1|\mathbf{x}) = \sigma(\mathbf{w}^\top \mathbf{x} + b)$. We calibrate scores by temperature scaling on a validation split ($\sigma(\cdot/T)$) and select the operating threshold by maximizing Youden's $J$ (sensitivity+specificity$-1$). All steps are batched and layer-aware, and the same pipeline is applied to encoder, decoder, and encoder–decoder models (and easily extends to masked sub-blocks and rectangular cross-attention).

**Experimental design.** We adopt an 85/15 training/held-out test protocol with hash-based splitting on the prompt. This avoids data leakage in HaluEval for example where we have two samples with the same prompt and a factual and a hallucinated answer. The training split is used for $k = 5$-fold cross-validation: in each fold we train the classifier and fit PCA on $k{-}1$ folds, then generate out-of-fold predictions on the held-out fold. Stacking all out-of-fold predictions, we first fit a temperature scaling parameter for score calibration and then select the operating threshold $\tau$ by maximizing Youden's $J$ statistic. Both calibration and threshold tuning are thus based entirely on training data. We then select the best model from the folds (including PCA), apply the fixed calibration transform, and evaluate on the unseen test set using the frozen threshold. Test-set metrics (AUROC, F1, Accuracy, ECE, and Brier score) are reported with 95% confidence intervals obtained from 1000 stratified bootstrap resamples, and paired bootstrap is used for significance testing between models.

**Datasets.** We evaluate on a paired subset of the HaluEval Li et al. (2023) question–answering split consisting of *10,000 prompts*, each with two model responses: one *correct* and one *hallucinated*. This yields a balanced binary classification setting with 20,000 instances (10,000 positive, 10,000 negative). Prompts cover open-domain QA; all items are single-turn question–response pairs. We also analyze a second dataset, TruthfulQA, which is a benchmark designed to test whether models avoid common human falsehoods and misconceptions. It contains 817 questions spanning 38 categories (e.g., health, law, finance, politics). Questions are written so that some humans would answer falsely; good performance requires resisting false but salient patterns from web text. The benchmark supports both *open-ended* generation and *multiple-choice* evaluation (MC1/MC2). Each question is accompanied by reference answers (true/false sets) used for automatic or human judging.

**Models.** In order to demonstrate the general utility of our approach we used GPT-2, a 137M parameter decoder model with 12 layers and 12 attention heads; Phi-2, a larger 2.7B parameter decoder model with 32 layers and 32 attention heads; and FLAN-T5 base a 248M parameter encoder–decoder model with 12 layers and 8 attention heads. While we analyze

the decoder self-attention matrices of GPT-2 and Phi-2, we are computing the bipartite cross operator $M$ to the cross attention demonstrating the ability to model rectangular matrices.

**Baselines.** We implemented two baselines to compare our approach to: token probability (TP) Quevedo et al. (2025) which computes statistics over the response token logits, and LapEigvals, which computes eigenvalues for each head in each layer, concatenates those and projects the vector into a 512-sized lower dimensional space and then into a logistic regression classifier Binkowski et al. (2025).

## 4.2 RESULTS

Table 1: Feature Set Complementarity Analysis Results over the HaluEval Dataset

| Model | Feature Set A | Feature Set B | RV Coeff. | 95% CI | p-value |
|-------|---------------|---------------|-----------|--------|---------|
| GPT-2 | LAPEIGVALS | SVD | 0.271 | [0.256, 0.294] | 0.002 |
| GPT-2 | EIGEN-FEATURES | SVD | 0.000 | [0.000, 0.000] | 0.16 |
| Phi-2 | EIGEN-FEATURES | SVD | 0.130 | [0.123, 0.136] | 0.002 |
| Phi-2 | LAPEIGVALS | SVD | 0.000 | [0.000, 0.000] | 1.00 |

RV Coefficient measures linear association between feature sets (0=independent, 1=identical).
CI = Confidence Interval computed via bootstrap sampling (n=1000).
p-values from permutation test under null hypothesis of independence.

**From representational relation (RV) to detection.** To quantify the relation between feature families, we compute the RV coefficient *after* projecting each family onto its own PCA subspace. This reduces scale/collinearity artifacts and yields a relation measure between *informative* linear subspaces rather than raw, high-dimensional coordinates. We report bootstrap 95% CIs (sequence/question-level resampling) and permutation $p$-values under the null of no linear association. We quantify the linear association between feature families using the RV coefficient Robert & Escoufier (1976). On **HaluEval** in table 1, except for the relation between LapEigvals and SVD features with Phi-2 we observe low to moderate potential complementarity. On **TruthfulQA** in the Appendix D in table 5, all relations except for Eigen-Features and SVD with the GPT-2 model show again low to moderate complementarity. Thus, the harder dataset elicits greater alignment between Eigen and SVD families, whereas LapEigVals stays only weakly coupled to SVD across models.

Table 2: SHADE Evaluation Results — Halueval

| Model | Feature Set | AUROC | Accuracy | F1 |
|-------|-------------|-------|----------|-----|
| T5 | TP | 0.789 ± 8.4e-03 | 0.489 ± 2.4e-03 | 0.045 ± 7.6e-03 |
| T5 | Cross SVD | **0.987 ± 2.2e-03** | **0.978 ± 2.7e-03** | **0.979 ± 2.6e-03** |
| GPT-2 | TP | 0.716 ± 9.6e-03 | 0.670 ± 8.9e-03 | 0.677 ± 9.6e-03 |
| GPT-2 | LapEigvals | 0.974 ± 3.1e-03 | 0.621 ± 6.5e-03 | 0.462 ± 1.4e-02 |
| GPT-2 | Eigen-Features | 0.492 ± 1.1e-02 | 0.510 ± 8.6e-03 | 0.613 ± 7.7e-03 |
| GPT-2 | SVD Coupling | **0.983 ± 2.4e-03** | **0.952 ± 4.0e-03** | **0.955 ± 3.8e-03** |
| Phi-2 | TP | 0.264 ± 9.7e-03 | 0.345 ± 6.5e-03 | 0.067 ± 7.8e-03 |
| Phi-2 | LapEigvals | 0.561 ± 5.2e-03 | 0.459 ± 1.1e-16 | 0.000 ± 0.0e+00 |
| Phi-2 | Eigen-Features | 0.985 ± 2.5e-03 | 0.971 ± 3.1e-03 | 0.973 ± 3.0e-03 |
| Phi-2 | SVD Coupling | **0.989 ± 2.0e-03** | **0.976 ± 2.8e-03** | **0.978 ± 2.7e-03** |

**Detection performance by model, dataset, and feature set.** Across all settings, the **SVD coupling** pathway is the most reliable detector. It is near-ceiling on **HaluEval** (see table 2 GPT-2 AUROC 0.983, Phi-2 0.989; FLAN-T5 cross-attention SVD 0.987) with strong accuracy/F1 and favorable calibration (low Brier/ECE see Appendix D in table 6).

Table 3: SHADE Evaluation Results — TruthfulQA

| Model | Feature Set | AUROC | Accuracy | F1 |
|-------|-------------|-------|----------|-----|
| T5 | TP | 0.482 ± 3.4e-02 | 0.500 ± 2.8e-02 | 0.410 ± 3.9e-02 |
| T5 | Cross SVD | **0.748 ± 2.9e-02** | **0.673 ± 2.8e-02** | **0.679 ± 2.7e-02** |
| GPT-2 | TP | 0.475 ± 3.5e-02 | 0.487 ± 3.8e-03 | 0.654 ± 1.7e-03 |
| GPT-2 | LapEigvals | 0.638 ± 3.3e-02 | 0.526 ± 2.0e-02 | 0.648 ± 1.5e-02 |
| GPT-2 | Eigen-Features | 0.723 ± 3.1e-02 | 0.642 ± 2.9e-02 | **0.662 ± 2.8e-02** |
| GPT-2 | SVD Coupling | **0.731 ± 3.1e-02** | **0.659 ± 2.9e-02** | 0.619 ± 3.5e-02 |
| Phi-2 | TP | 0.505 ± 3.4e-02 | 0.516 ± 1.1e-16 | 0.000 ± 0.0e+00 |
| Phi-2 | LapEigvals | 0.497 ± 2.1e-02 | 0.512 ± 3.7e-03 | 0.000 ± 0.0e+00 |
| Phi-2 | Eigen-Features | 0.764 ± 2.9e-02 | 0.672 ± 2.7e-02 | 0.617 ± 3.6e-02 |
| Phi-2 | SVD Coupling | **0.825 ± 2.5e-02** | **0.708 ± 2.8e-02** | **0.677 ± 3.3e-02** |

On **TruthfulQA** (table 3), SVD remains best or tied (GPT-2 0.731, Phi-2 0.825; FLAN-T5 cross-attention 0.748), with competitive calibration. **LapEigvals** shows mixed behavior: it can rank reasonably on HaluEval (GPT-2 AUROC 0.974) but yields weak thresholded metrics and poorer calibration, and it degrades on TruthfulQA (GPT-2 0.638, Phi-2 $\approx$ 0.50). **Eigen-Features** are variable—close to SVD on some pairs (Phi-2/HaluEval 0.985), yet unstable on others (GPT-2/HaluEval $\approx$ 0.49). **Token-probability** baselines exhibit consistently lower discrimination and do not improve calibration in our setup.

## 5 CONCLUSION

We presented **SHADE**, a unified framework for effective hallucination detection based on analyzing equilibrium structural information through dual spectral perspectives: directed Laplacian analysis (information flow) and bipartite SVD (query–key coupling). Our contributions span both conceptual framing and empirical validation.

**Conceptual Contributions.** We proposed that flow- and coupling-based features capture distinct structural failure modes. This hypothesis builds on spectral intuition: directed Laplacians emphasize global flow bottlenecks, while bipartite SVD highlights localized coupling between queries and keys. Unlike LapEigVals, which is restricted to square matrices, our bipartite SVD formulation applies to all transformer architectures, including those with rectangular attention, enabling broad applicability.

**Empirical Validation.** We measured representational relations between feature families and found low RV coefficients, supporting the view that flow and coupling perspectives extract complementary information. Each spectral perspective alone achieves strong detection performance, with SVD coupling consistently outperforming all other baselines. Eigen features yield more mixed results: they approach SVD performance in some settings (e.g., Phi-2 on HaluEval) but lag substantially in others (e.g., GPT-2 on HaluEval). LapEigvals features show moderate effectiveness depending on the model and dataset. Taken together with our representational relation analysis (low RV coefficients), these results support the view that flow- and coupling-based features capture complementary structural information even though we have not yet explicitly combined them in a joint detector.

**Future Work.** Promising directions include sparse approximations for long contexts, physics-inspired zero-shot spectral lenses, concrete explainable insights, and adversarial robustness of spectral signatures. Further, we are interested to investigate techniques to integrate the value geometry that we ignored in this analysis.

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

# A    Understanding Out-Degree Computation in LapEigvals

This section clarifies LapEigvals' convention for attention matrices and their out-degree computation for hallucination detection.

## A.1    Notation Convention

Following LapEigvals, the attention matrix $A$ uses the convention where $A_{ij}$ represents attention that token $i$ RECEIVES FROM token $j$ (edge $j \rightarrow i$). This means:

- **Row** $i$: Shows where token $i$ gets its attention from
- **Column** $j$: Shows where token $j$ sends its attention to
- **Row sums**: Always 1 (in-degree, by softmax normalization)
- **Column sums**: Vary (out-degree, total influence on other tokens)

For the out-degree matrix:

- **Out-degree of token** $j$: $(D_{\text{out}})_{jj} = \frac{\sum_{i>j} A_{ij}}{|i>j|}$
- Sum entries below diagonal in column $j$ (future tokens receiving from $j$)
- Normalize by count of future tokens

## A.2    Visual Illustration

Figure 2 illustrates the computation for a 5-token sequence.

**Attention Matrix and Out-Degree Computation (LapEigvals)**

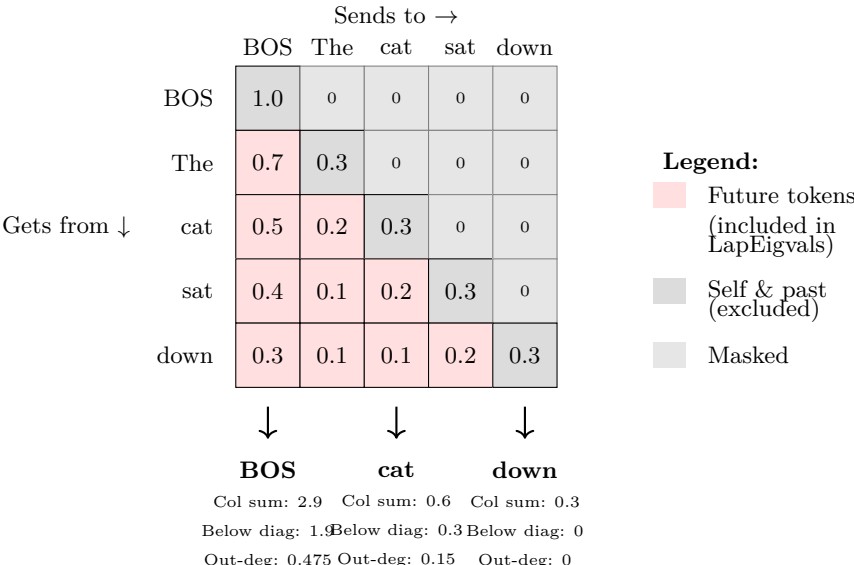

Figure 2: Out-degree computation following LapEigvals' convention. Matrix element $A_{ij}$ shows attention token $i$ (row) receives FROM token $j$ (column). Red cells below the diagonal in each column are summed for out-degree, these show future tokens receiving attention from that column's token. Gray cells (diagonal and above) are excluded. The normalized out-degree measures how much each token influences future predictions.

## A.3 Detailed Computation Example

With LapEigvals' convention where $A_{ij}$ = attention token $i$ receives FROM token $j$:

### A.3.1 BOS (Column 0)

**Full column sum:**

$$\text{Column sum}_{\text{BOS}} = \sum_{i=0}^{4} A_{i,0} \tag{5}$$
$$= A_{0,0} + A_{1,0} + A_{2,0} + A_{3,0} + A_{4,0} \tag{6}$$
$$= 1.0 + 0.7 + 0.5 + 0.4 + 0.3 = 2.9 \tag{7}$$

**LapEigvals out-degree (below diagonal only):**

$$\text{Below diagonal}_{\text{BOS}} = \sum_{i>0} A_{i,0} = 0.7 + 0.5 + 0.4 + 0.3 = 1.9 \tag{8}$$
$$\text{Future count}_{\text{BOS}} = |\{i : i > 0\}| = 4 \tag{9}$$
$$D_{\text{out},00} = \frac{1.9}{4} = 0.475 \tag{10}$$

### A.3.2 cat (Column 2)

**Full column sum:**

$$\text{Column sum}_{\text{cat}} = \sum_{i=0}^{4} A_{i,2} \tag{11}$$
$$= 0 + 0 + 0.3 + 0.2 + 0.1 = 0.6 \tag{12}$$

**LapEigvals out-degree (below diagonal only):**

$$\text{Below diagonal}_{\text{cat}} = \sum_{i>2} A_{i,2} = 0.2 + 0.1 = 0.3 \tag{13}$$
$$\text{Future count}_{\text{cat}} = |\{i : i > 2\}| = 2 \tag{14}$$
$$D_{\text{out},22} = \frac{0.3}{2} = 0.15 \tag{15}$$

### A.3.3 down (Column 4)

**Full column sum:**

$$\text{Column sum}_{\text{down}} = \sum_{i=0}^{4} A_{i,4} = 0.3 \tag{16}$$

**LapEigvals out-degree (below diagonal only):**

$$\text{Below diagonal}_{\text{down}} = \sum_{i>4} A_{i,4} = 0 \text{ (no future tokens)} \tag{17}$$
$$D_{\text{out},44} = 0 \tag{18}$$

## A.4 Interpretation for Hallucination Detection

The normalized out-degree captures how much each token influences future predictions:

- **High out-degree** (e.g., BOS with 0.475): Token acts as an information hub that future tokens heavily depend on

- **Low out-degree** (e.g., later tokens): Less influence on future, expected due to fewer future tokens

- **Anomalous patterns**: Unexpected drops in out-degree may indicate the model is ignoring important context

Why this detects hallucinations:

1. **Normal pattern**: Early tokens (especially BOS) maintain high out-degree as future tokens reference them

2. **Hallucination pattern**: Out-degree drops abnormally when the model "forgets" context and generates freely

3. **Position normalization**: Dividing by future token count removes trivial position bias

The key insight: When a model hallucinates, future tokens stop depending on earlier context tokens, manifesting as abnormally low out-degree values in the spectral analysis.

# B  Mathematical Analysis of LapEigvals' Spectral Limitations

This appendix provides a detailed mathematical analysis of the fundamental limitations of the LapEigvals approach Binkowski et al. (2025), contrasting it with standard spectral graph theory to highlight why SHADE's dual decomposition offers superior theoretical guarantees.

## B.1  Setup and Construction

For causal self-attention $A \in \mathbb{R}^{n \times n}$ where rows represent queries and columns represent keys, with softmax normalization ensuring row sums equal 1 and lower-triangular structure (strictly causal attention), LapEigvals defines:

$$L = D - A \tag{19}$$

$$D_{ii} = \frac{\sum_{j>i} A_{ji}}{|\{j : j > i\}|} \tag{20}$$

where $D$ is a diagonal "future out-degree" matrix counting column sums below the diagonal, normalized by the number of future tokens. Because $A$ is lower-triangular and $D$ is diagonal, $L$ is lower-triangular and its eigenvalues are the diagonal entries:

$$\lambda_i(L) = D_{ii} - A_{ii}$$

The method reads these eigenvalues directly without eigenvalue decomposition.

## B.2  Fundamental Spectral Limitations

### B.2.1  Non-PSD, Non-Hermitian Structure

The matrix $L = D - A$ is generally **non-symmetric** and **not positive semi-definite**. This eliminates classical spectral graph guarantees:

**Theorem B.1** (Absence of Spectral Guarantees). *The LapEigvals construction $L = D - A$ lacks the mathematical structure required for standard spectral graph theory:*

1. *$L$ is not symmetric: $(L)_{ij} \neq (L)_{ji}$ in general*

2. *$L$ is not PSD: $\langle x, Lx \rangle$ can be negative for some $x \in \mathbb{R}^n$*

3. *Cheeger-type inequalities do not apply*

4. *Mixing time interpretations are unavailable*

*Proof.* For the lower-triangular attention matrix $A$, we have $A_{ij} = 0$ for $j > i$. The Laplacian $L = D - A$ has:

$$L_{ij} = \begin{cases} D_{ii} - A_{ii} & \text{if } i = j \\ -A_{ij} & \text{if } i \neq j \end{cases} \tag{21}$$

Since $A_{ij} \neq A_{ji}$ in general (only upper triangle is zero), $L$ is non-symmetric. The lack of positive semi-definiteness follows from the arbitrary signs in the off-diagonal entries and the dependence of $D_{ii}$ on future positions. $\square$

In contrast, Chung's directed Laplacian Chung (2005) constructs a symmetric operator with real spectrum and established inequalities linking eigenvalues to expansion properties.

### B.2.2 Eigenvalue Collapse to Diagonal Entries

**Theorem B.2** (Spectral Information Collapse). *For the triangular construction $L = D - A$, the eigenvalues $\lambda_i(L) = D_{ii} - A_{ii}$ depend only on:*

1. *The self-loop strength $A_{ii}$ (local recurrence)*

2. *The future out-degree $D_{ii}$ (how much future tokens attend to token $i$)*

*All off-diagonal coupling patterns in the sub-diagonal entries are ignored by the spectrum.*

*Proof.* Since $L$ is lower-triangular, its eigenvalues are the diagonal entries Horn & Johnson. The coupling between different tokens through $A_{ij}$ for $i \neq j$ affects the eigenvectors but not the eigenvalues. Community structure, clustering patterns, and inter-token dependencies encoded in the off-diagonal elements are invisible to the spectral analysis. $\square$

This represents a fundamental information loss compared to full spectral decomposition, where off-diagonal structure determines the eigenspace geometry and community detection properties von Luxburg (2007).

### B.2.3 Position and Length Dependence

**Theorem B.3** (Non-Permutation Invariance). *The degree matrix $D$ defined in Equation 20 is not permutation-invariant:*

1. *$D_{ii}$ depends on absolute position $i$ in the sequence*

2. *The count $|\{j : j > i\}|$ depends on sequence length*

3. *Graph isomorphisms that preserve attention patterns but change indexing yield different "eigenvalues"*

*Proof.* Consider two attention matrices $A$ and $A'$ that are permutation-equivalent: $A' = PAP^T$ for some permutation matrix $P$. The corresponding degree matrices $D$ and $D'$ will differ because the future out-degree calculation depends on the specific indices. For token at position $i$, moving it to position $i'$ changes both the numerator $\sum_{j>i'} A'_{ji'}$ and denominator $|\{j : j > i'\}|$. $\square$

Standard graph Laplacians are permutation-invariant, making their eigenvalues intrinsic graph properties independent of node labeling Godsil & Royle (2001).

### B.2.4 Under-utilized Markov Structure

The row-stochastic property of $A$ (rows sum to 1) enables natural random-walk interpretations, but LapEigvals' construction $D$ does not correspond to the standard Markov generator.

**Theorem B.4** (Lost Random-Walk Interpretation)**.** *For the random-walk Laplacian $L_{rw} = I - A$, eigenvalues have clear probabilistic meaning:*

$$\lambda_i(L_{rw}) = 1 - \lambda_i(A) \tag{22}$$

*where $\lambda_i(A)$ relates to contraction rates of the Markov process. LapEigvals' construction loses this connection.*

The random-walk Laplacian provides spectral gaps linked to mixing times and conductance Levin et al. (2006); Chung (1997), while LapEigvals' eigenvalues lack standard probabilistic interpretation.

### B.2.5 Causality Restriction

**Theorem B.5** (Non-Portability)**.** *The triangular eigenvalue reading $\lambda_i(L) = D_{ii} - A_{ii}$ breaks for:*

*1. Cross-attention matrices (rectangular, non-square)*

*2. Bidirectional attention (non-triangular)*

*3. Any attention pattern where $A_{ij} \neq 0$ for $j > i$*

*Without triangular structure, the method loses its computational advantage while lacking guarantees of standard Laplacian constructions.*

## B.3 Standard Spectral Graph Alternatives

### B.3.1 Random-Walk Laplacian

For row-stochastic $A$, the random-walk Laplacian $L_{rw} = I - A$ provides:

$$\text{Spectrum:} \quad \lambda_i(L_{rw}) = 1 - \lambda_i(A) \tag{23}$$
$$\text{Interpretation:} \quad \text{Spectral gap} \leftrightarrow \text{Mixing time} \tag{24}$$
$$\text{Bounds:} \quad \rho(A) \leq 1 \text{ with equality iff reducible} \tag{25}$$

For triangular $A$, this still yields $\lambda_i(L_{rw}) = 1 - A_{ii}$ with clear "self-leakiness" interpretation Lovász (1996).

### B.3.2 Chung's Directed Laplacian

Given transition matrix $P$ and stationary distribution $\phi$, Chung's construction Chung (2005):

$$L_{\text{Chung}} = I - \frac{1}{2}\left(\Phi^{1/2}P\Phi^{-1/2} + \Phi^{-1/2}P^T\Phi^{1/2}\right) \tag{26}$$

provides real spectrum and directed Cheeger inequalities Cheeger (2015); Sinclair & Jerrum. For doubly stochastic matrices, this reduces to the symmetric operator $L_{\text{Chung}} = I - \frac{1}{2}(P + P^T)$.

## B.4 Summary Comparison

The analysis reveals that while LapEigvals offers computational convenience for strictly causal attention, it sacrifices fundamental mathematical guarantees and interpretability available in standard spectral graph theory. This motivates SHADE's dual decomposition approach, which preserves spectral structure while extending to general attention architectures.

# C Mathematical Note: Bipartite Laplacian and SVD Correspondence

While our main results focus on the complementarity between directed Laplacian flow analysis and bipartite SVD coupling analysis, we include here an interesting mathematical relationship for completeness.

| Property | LapEigvals | Random-Walk | Chung |
|---|---|---|---|
| Symmetric | ✗ | ✗ | ✓ |
| Real Spectrum | ✓ | Depends | ✓ |
| PSD Guarantees | ✗ | ✗ | ✓ |
| Cheeger Bounds | ✗ | Limited | ✓ |
| Markov Interpretation | ✗ | ✓ | ✓ |
| Permutation Invariant | ✗ | ✓ | ✓ |
| Cross-Attention | ✗ | ✓ | ✓ |

Table 4: Comparison of spectral properties across Laplacian constructions.

## C.1 Graph-Theoretic Interpretation

Treating $(Q, K)$ as a bipartite graph with weights $A_{ij}$, our $M$ is the **normalized biadjacency**. The symmetric block operator

$$\mathcal{B} = \begin{pmatrix} 0 & M \\ M^T & 0 \end{pmatrix} \tag{27}$$

has eigenvalues $\{\pm\sigma_r(M)\}$, so standard spectral graph notions (gaps, conductance-like statements, clustering) apply naturally. This neatly generalizes directed Laplacian ideas without committing to square/triangular structure, enabling analysis of cross-attention where traditional methods fail.

## C.2 The Deterministic Correspondence

When we construct a bipartite Laplacian from the same normalized cross-operator $M$ that we analyze via SVD, there exists a deterministic relationship between the eigenvalues and singular values:

**Theorem C.1** (Hermitian dilation / bipartite correspondence). *Let $M = D_Q^{\dagger 1/2} B D_K^{\dagger 1/2}$ be the normalized cross-operator (rectangular allowed). Define the bipartite adjacency $H(M) = \begin{bmatrix} 0 & M \\ M^\top & 0 \end{bmatrix}$ and Laplacian $\mathcal{L}_{bip} = I - H(M)$. Then the eigenvalues of $H(M)$ are the signed singular values $\{\pm\sigma_i(M)\}$ (with extra zeros if $M$ is rectangular). Consequently, $\mathcal{L}_{bip}$ has eigenvalues $1 \pm \sigma_i(M)$ plus 1 with multiplicity $n_q + n_k - 2r$, where $r = \text{rank}(M)$.*

*Proof sketch.* If $Mv_i = \sigma_i u_i$ and $M^\top u_i = \sigma_i v_i$, then $H(M)[u_i; \pm v_i] = \pm\sigma_i[u_i; \pm v_i]$. □

**Corollary C.2** (Doubly-stochastic special case). *If $B$ is square and doubly-stochastic, then $M = B$, the top singular value is $\sigma_1 = 1$ with $u_1 = v_1 = \mathbf{1}/\sqrt{n}$, and $\mathcal{L}_{bip}$ has eigenvalues 0 and 2 corresponding to $1 \pm \sigma_1$; all remaining eigenvalues lie in $(0, 2)$.*

## C.3 Computational Implications

This correspondence means that if we were to use eigendecomposition of the bipartite Laplacian, we would obtain the same information as SVD of $M$, just transformed via $\lambda = 1 \pm \sigma$. Therefore:

1. **Computational efficiency**: SVD of $M$ (an $n_q \times n_k$ matrix) is more efficient than eigendecomposition of $\mathcal{L}_{bip}$ (an $(n_q + n_k) \times (n_q + n_k)$ matrix)

2. **Direct interpretation**: Singular values directly measure coupling strength, while eigenvalues require the transformation $\sigma = |1 - \lambda|$

3. **Numerical stability**: SVD algorithms are typically more stable for rectangular matrices

# D Additional Experimental Results

Table 5: Full Feature Set Complementarity Analysis Results

| Model | Dataset | Feature Set A | Feature Set B | RV Coeff. | 95% CI | p-value |
|---|---|---|---|---|---|---|
| GPT-2 | HaluEval | LapEigVals | SVD | 0.271 | [0.256, 0.294] | 0.002 |
| GPT-2 | HaluEval | Eigen-Features | SVD | 0.000 | [0.000, 0.000] | 0.16 |
| GPT-2 | TruthfulQA | Eigen-Features | SVD | 0.944 | [0.938, 0.947] | 0.002 |
| GPT-2 | TruthfulQA | LapEigVals | SVD | 0.408 | [0.377, 0.446] | 0.002 |
| Phi-2 | HaluEval | Eigen-Features | SVD | 0.130 | [0.123, 0.136] | 0.002 |
| Phi-2 | HaluEval | LapEigVals | SVD | 0.000 | [0.000, 0.000] | 1.00 |
| Phi-2 | TruthfulQA | Eigen-Features | SVD | 0.771 | [0.737, 0.802] | 0.002 |
| Phi-2 | TruthfulQA | LapEigVals | SVD | 0.011 | [0.009, 0.015] | 0.002 |

RV Coefficient measures linear association between feature sets (0=independent, 1=identical).
CI = Confidence Interval computed via bootstrap sampling (n=1000).
p-values from permutation test under null hypothesis of independence.

Table 6: SHADE Evaluation Results — All Experiments

| Model | Dataset | Feature Set | AUROC | Accuracy | F1 | ECE | Brier |
|---|---|---|---|---|---|---|---|
| T5 | TruthfulQA | TP | 0.482 ± 3.4e-02 | 0.500 ± 2.8e-02 | 0.410 ± 3.9e-02 | 0.111 ± 2.3e-02 | 0.261 ± 5.1e-03 |
| T5 | TruthfulQA | Cross SVD | **0.748 ± 2.9e-02** | **0.673 ± 2.8e-02** | **0.679 ± 2.7e-02** | **0.109 ± 2.1e-02** | **0.203 ± 1.3e-02** |
| T5 | HaluEval | TP | 0.789 ± 8.4e-03 | 0.489 ± 2.4e-03 | 0.045 ± 7.6e-03 | 0.206 ± 6.6e-03 | 0.237 ± 1.5e-03 |
| T5 | HaluEval | Cross SVD | **0.987 ± 2.2e-03** | **0.978 ± 2.7e-03** | **0.979 ± 2.6e-03** | **0.014 ± 2.2e-03** | **0.021 ± 2.3e-03** |
| GPT-2 | HaluEval | TP | 0.716 ± 9.6e-03 | 0.670 ± 8.9e-03 | 0.677 ± 9.6e-03 | 0.084 ± 8.4e-03 | 0.218 ± 2.7e-03 |
| GPT-2 | HaluEval | LapEigvals | 0.974 ± 3.1e-03 | 0.621 ± 6.5e-03 | 0.462 ± 1.4e-02 | 0.299 ± 4.6e-03 | 0.153 ± 1.2e-03 |
| GPT-2 | HaluEval | Eigen-Features | 0.492 ± 1.1e-02 | 0.510 ± 8.6e-03 | 0.613 ± 7.7e-03 | **0.002 ± 6.5e-04** | 0.248 ± 1.5e-04 |
| GPT-2 | HaluEval | SVD Coupling | **0.983 ± 2.4e-03** | **0.952 ± 4.0e-03** | **0.955 ± 3.8e-03** | 0.022 ± 3.1e-03 | **0.039 ± 2.6e-03** |
| GPT-2 | TruthfulQA | TP | 0.475 ± 3.5e-02 | 0.487 ± 3.8e-03 | 0.654 ± 1.7e-03 | 0.370 ± 5.3e-03 | 0.391 ± 5.5e-03 |
| GPT-2 | TruthfulQA | LapEigvals | 0.638 ± 3.3e-02 | 0.526 ± 2.0e-02 | 0.648 ± 1.5e-02 | **0.083 ± 2.6e-02** | 0.245 ± 1.5e-03 |
| GPT-2 | TruthfulQA | Eigen-Features | 0.723 ± 3.1e-02 | 0.642 ± 2.9e-02 | **0.662 ± 2.8e-02** | 0.101 ± 2.3e-02 | 0.214 ± 1.2e-02 |
| GPT-2 | TruthfulQA | SVD Coupling | **0.731 ± 3.1e-02** | **0.659 ± 2.9e-02** | 0.619 ± 3.5e-02 | 0.107 ± 2.2e-02 | **0.211 ± 1.2e-02** |
| Phi-2 | HaluEval | TP | 0.264 ± 9.7e-03 | 0.345 ± 6.5e-03 | 0.067 ± 7.8e-03 | 0.215 ± 8.3e-03 | 0.292 ± 1.7e-03 |
| Phi-2 | HaluEval | LapEigvals | 0.561 ± 5.2e-03 | 0.459 ± 1.1e-16 | 0.000 ± 0.0e+00 | 0.019 ± 2.4e-06 | 0.249 ± 2.4e-06 |
| Phi-2 | HaluEval | Eigen-Features | 0.985 ± 2.5e-03 | 0.971 ± 3.1e-03 | 0.973 ± 3.0e-03 | 0.023 ± 2.8e-03 | 0.026 ± 2.4e-03 |
| Phi-2 | HaluEval | SVD Coupling | **0.989 ± 2.0e-03** | **0.976 ± 2.8e-03** | **0.978 ± 2.7e-03** | **0.018 ± 2.6e-03** | **0.021 ± 2.3e-03** |
| Phi-2 | TruthfulQA | TP | 0.505 ± 3.4e-02 | 0.516 ± 1.1e-16 | 0.000 ± 0.0e+00 | 0.475 ± 2.8e-03 | 0.471 ± 3.3e-03 |
| Phi-2 | TruthfulQA | LapEigvals | 0.497 ± 2.1e-02 | 0.512 ± 3.7e-03 | 0.000 ± 0.0e+00 | **0.004 ± 1.3e-05** | 0.250 ± 1.3e-05 |
| Phi-2 | TruthfulQA | Eigen-Features | 0.764 ± 2.9e-02 | 0.672 ± 2.7e-02 | 0.617 ± 3.6e-02 | 0.080 ± 1.9e-02 | 0.197 ± 1.1e-02 |
| Phi-2 | TruthfulQA | SVD Coupling | **0.825 ± 2.5e-02** | **0.708 ± 2.8e-02** | **0.677 ± 3.3e-02** | 0.095 ± 2.0e-02 | **0.172 ± 1.3e-02** |

