# OpenReview forum: "SHADE: Spectral Hallucination Detection via Dual Spectral Decompositions"
_ICLR.cc/2026/Conference — Submitted to ICLR 2026_

### Official Review · Reviewer_g6mG · 2025-10-28

**Soundness:** 3
**Presentation:** 1
**Contribution:** 2
**Rating:** 2
**Confidence:** 3

**Summary:**

This paper introduces SHADE, a novel framework for detecting hallucinations in Large Language Models by analyzing the internal structure of their attention mechanisms. The method uniquely employs a dual-view approach to diagnose distinct failure modes.

The first view analyzes the information flow within the attention network, identifying when the model detaches from the source prompt and begins to recycle its own generated content. The second view assesses the query-key coupling, measuring the alignment strength between what the model is looking for and the information it finds. This detects brittle connections, such as when attention collapses onto a few non-informative tokens.

The authors demonstrate that these two perspectives are complementary and capture different types of structural failures. Through extensive experiments on various models and benchmarks, SHADE is shown to significantly outperform previous detection methods. The framework is notable for being theoretically grounded, applicable across diverse model architectures, and providing interpretable signals for why a model is hallucinating.

**Strengths:**

1. Addresses a Timely and Critical Problem: The authors tackle the problem of hallucination in Large Language Models, which remains a significant and pressing challenge for the field. The work's focus on developing principled, internal methods for detecting such failures is highly relevant and contributes to a crucial line of research aimed at improving model safety and reliability.

2. Conceptually Interesting Premise: The core intuition to diagnose attention failures by modeling the mechanism from two distinct structural viewpoints—information flow and query-key coupling—is conceptually novel and interesting. This dual-perspective approach offers a potentially more holistic way to interpret the internal state of the model compared to methods that rely on a single analytical lens.

**Weaknesses:**

1. The core weakness is the inherent ambiguity of its spectral features, which are incapable of distinguishing between attention patterns essential for correct reasoning and pathological ones that lead to hallucinations. For instance, a strong, necessary focus on a key entity to answer a factual question produces the same "over-concentration" signal that the method flags as high-risk. Similarly, a simple and correct copying mechanism, often used in summarization, results in a "rank collapse", a pattern the method incorrectly associates with pathological over-fitting. Because these features are fundamentally context-blind—describing only the geometric shape of the attention distribution rather than its semantic appropriateness for the task—they cannot serve as reliable, standalone indicators of hallucination.

2. A significant number of hallucinations emerge dynamically over the course of generation. Therefore, for complex outputs, a methodology that relies exclusively on a static analysis of the final attention distribution is theoretically insufficient for accurate detection. Consider a "snowball effect," where an initial, single-step hallucination leads to a cascade of subsequent errors. Given the final length of the text, the original trigger for this failure mode would likely be obscured and thus undetectable by a post-hoc static analysis. This scenario is particularly critical as it more closely mirrors real-world failure cases.

**Questions:**

1. I am skeptical about the practical viability of using attention features as a primary mechanism for hallucination detection. My main concern stems from the significant computational overhead required to access and process attention distributions during the inference phase. For each token generated, this method necessitates retrieving the attention scores from multiple layers and heads, which is a computationally expensive operation. This introduces non-trivial latency and reduces throughput, especially when compared to standard, optimized inference processes that do not require inspecting these intermediate states. Does the potential gain in detection accuracy justify the substantial performance cost? In many application scenarios, such as real-time chatbots or large-scale content generation, inference speed is a paramount constraint. Therefore, a method that significantly slows down generation may not be a feasible solution, regardless of its theoretical effectiveness.

2. A reliance on the final static attention distribution for detection, without the capacity to integrate the temporal dynamics of attention features in long sequences, raises a critical challenge: how can such an approach detect the "snowball effect" originating from a single-step hallucination within a long text? This class of cascading hallucination represents a more prevalent and practical scenario, yet the proposed methodology appears to completely circumvent the handling of this realistic failure mode.

---

> ### Author Response · Authors · 2025-11-15
>
> We thank the reviewer for the detailed and constructive feedback. We agree that
> the presentation can be improved and will revise the paper to make our narrative
> clearer: **spectral features *are* effective, but only when built on
> mathematically sound operators.**
>
> **1. On "ambiguity" of spectral features.**
>
> We agree that raw attention "over-concentration" is not inherently pathological.
> The ambiguity highlighted by the reviewer arises predominantly from *operator
> choice*. Prior work (LapEigvals) uses an out-degree Laplacian whose spectrum
> collapses to diagonal terms and is highly position/length-dependent, discarding
> global structure (Appendix A–B) . When this operator is swapped for (i) the
> **random-walk Laplacian** (L_{rw}=I-A) and (ii) the **degree-normalized
> cross-operator** (M=D_Q^{-1/2}AD_K^{-1/2}), the resulting spectral geometry is
> stable, interpretable, and empirically robust.
>
> This is visible in our results: **Eigen-Features** (using a proper Laplacian)
> succeed where LapEigvals collapses (e.g., Phi-2/HaluEval: 0.985 AUROC vs.
> ≈0.50), and **SVD coupling** is consistently strongest (GPT-2/HaluEval 0.983;
> Phi-2/HaluEval 0.989; GPT-2/TruthfulQA 0.731; Phi-2/TruthfulQA 0.825; Tables
> 2–3) . Importantly, the cross-operator disambiguates healthy vs. pathological
> concentration:
>
> * **Healthy focus** → strong (\sigma_1) *with* multi-mode support (higher effective rank).
> * **Brittle anchoring** → rank-1 collapse and large spectral gap.
> * **Prompt detachment** → weak early coupling ((\sigma_1,\sigma_2)) and low flow-leakiness.
>
> These distinctions come directly from the operator semantics (Sec. 3; Fig. 1) .
>
> **2. On static vs. dynamic detection ("snowball effect").**
>
> Although our experiments report features on the final sequence, the method
> itself is **not** static. As described in Section4.1, SHADE records **per-layer,
> per-head attention during inference** in a fully batched fashion . This allows
> **prefix-wise or sliding-window diagnostics**: the same features can be computed
> on growing prefixes, enabling early detection of cascading hallucinations via
> declines in (\sigma_1), collapse of coupling entropy, or shrinking
> flow-leakiness. We will make this explicit by adding a short "Streaming SHADE"
> clarification in the revision.
>
> **3. Practical feasibility (latency and overhead).**
>
> We emphasize that SHADE uses **attention matrices already produced by the
> model**. Extracting them is zero extra compute; feature computation is
> lightweight:
>
> * A transformer with **L layers** and **H heads** produces **L×H** attention
>   maps.
> * For **top-2 SVD**, each map requires just a few power iterations on an
>   (n\times n) (or (n_q\times n_k)) matrix.
> * This is (O(LH \cdot n^2)) *with small constants*, and in streaming mode can be
>   applied to a **rolling window** of size (w\ll n), making the cost effectively
>   (O(LH \cdot w^2)).
> * Flow-based features (trace, entropy) are (O(n)).
>
> These costs are modest relative to the transformer's own attention computation,
> and because features are independent across heads, the implementation is fully
> parallelizable. At deployment time, SHADE can be run in a streaming manner
> because attention rows are produced incrementally and are compatible with
> standard k-v caching. In this setting, top-1/top-2 coupling modes can be
> maintained with lightweight matrix–vector power-iteration updates, avoiding full
> SVDs and keeping per-token overhead negligible. We estimate that with these runtime
> improvements in computation we are dealing with a low single digit percentage overhead
> in FLOPS.
>
> We will add a brief runtime note summarizing this argument.
>
> **4. Clarifying the choice of spectral resolution (k).**
>
> As described in Section4.1, we select (k=n_{\min}) to retain ≥90% of sequences
> (empirically **(k=17) for HaluEval** and **(k=14) for TruthfulQA**) . This
> avoids dataset-induced truncation while keeping SVDs small.
>
> Critically, however, **most discriminative signal lies in the top 1–2 singular
> values** ((\sigma_1,\sigma_2)) and a single flow scalar. This is consistent with
> the operator semantics: brittle anchoring is captured by the top singular mode
> and its gap, while prompt detachment is captured by early coupling energy + flow
> leakiness. This insight underpins our planned **zero-shot, top-2 SVD detector**.
>
> **5. Future work enabled by this paper.**
>
> As the reviewer notes, context-aware interpretation is essential. Our analysis
> deliberately evaluates full spectral observables to assess viability, and it
> paves the way for two targeted follow-ups:
>
> 1. **Zero-shot detection using top-2 SVD + one flow feature.**
>    A small, architecture-agnostic risk score using ({\sigma_1,\sigma_2}) +
>    leakiness, designed for streaming deployment.
>
> 2. **Spectrally informed mitigation.**
>    Training-time objectives that penalize brittle coupling (rank-1 collapse) or
>    insufficient flow leakiness—directly targeting the two structural failure
>    modes surfaced by our operators.

---

### Official Review · Reviewer_WziU · 2025-10-30

**Soundness:** 2
**Presentation:** 1
**Contribution:** 2
**Rating:** 2
**Confidence:** 4

**Summary:**

If I'm understanding correctly- and this paper has a lot of (what I think is) unnecessary jargon, so this has been difficult-

The core idea this paper proposes for detecting hallucinations is essentially to use the transformer's attention matrices to compute spectral statistics (eigenvalues, singular values, entropies, etc.) and then feed these handcrafted feature numbers into logistic regression to classify whether the output is a hallucination or not.

They test a series of models like GPT2/Phi2 on HaluEval and TruthfulQA (which are quite different datasets, given that HaluEval is looking at in-context grounded hallucination while TruthfulQA is moreso factuality without checking against a grounding source), and compare their handcrafted feature logistic regression model against token probability detectors and LapEigvals (a similar spectral baseline that uses a triangular Laplacian). TLDR, the authors method outperforms all them both, though the improvement is something like 1% absolute improvement against LapEigvals on most comparisons.

The core claim seems to be that simple logistic regression on spectral features from attention matrices detects hallucinations better than token-probability baselines.

**Strengths:**

The empirical finding seems relatively solid- that SVD-based coupling features consistently outperform token-probability and older spectral baselines. This seems like a relatively lightweight pipeline.

**Weaknesses:**

The main issue with this paper is that it tries to overcomplicate what is essentially a very simple approach. It buries this under heavy and unnecessary math, but it's really just simple linear algebra.

The baselines comparisons aren't exactly convincing- many, many hallucination detection methods have been proposed, like semantic entropy, self-consistency, embedding-based factuality, surface based classification, etc. Comparing against only 2 baselines with no justification as to why they were selected seems like cherry picking.

Furthermore, there is little analysis on cross-modal or cross-dataset generalization. the logistic regression classifiers are trained per model and per dataset; how extensive this extends to other models and domains remains a question- it can be very convincingly argued that token-probability (uncertainty-based) metrics are much more generalizable across domains and models, and more lightweight to compute, than needing to train the authors' approach across every single model and domain.

**Questions:**

NA

---

> ### Author Response · Authors · 2025-11-15
> **Spectral features work, but operator choice matters**
>
> We thank the reviewer for their detailed feedback. Our core clarification is:
>
> > **Spectral features work, but operator choice determines whether they remain
>     meaningful and reliable at scale.**
>
> Below we address each concern directly.
>
> ---
>
> ## **1. "Overcomplicated for a simple idea."**
>
> We agree the *classifier* is simple—by design. However, the *operator* behind
> each spectral feature is what determines signal quality. LapEigvals demonstrated
> that eigenvalue-based features can detect hallucinations, but the **out-degree
> Laplacian** it uses has fundamental limitations (non-PSD, collapses to diagonal
> entries, position-dependent, and inapplicable to cross-attention). These issues
> explain why it works on small models but breaks on larger ones.
>
> Our contribution is to use **mathematically sound operators**:
>
> * **Random-walk Laplacian** → restores proper Markov semantics and permutation
>     stability; **Eigen-Features regain strong performance** (Phi-2/HaluEval
>     0.985; TruthfulQA 0.764).
>
> * **Degree-normalized cross-operator** (M = D_Q^{-1/2} A D_K^{-1/2}) →
>     architecture-agnostic, handles rectangular attention, and yields physically
>     meaningful SVD metrics; **consistently best** (GPT-2/HaluEval 0.983;
>     Phi-2/HaluEval 0.989; T5 0.987).
>
> Thus, the main insight is not "more linear algebra," but identifying the
> **correct operators** that make spectral features reliable.
>
>
> ## **2. "Baselines are limited."**
>
> Our scope is **single-pass internal detection**, so token-probability and
> LapEigvals are the appropriate canonical comparisons.
>
> The progression of results directly supports our operator argument:
>
> * **LapEigvals**: good on GPT-2/HaluEval (0.974) but *collapses* for Phi-2
>     (≈0.50) because its operator destroys signal under PCA.
> * **Eigen-Features**: strong once the operator is fixed.
> * **SVD cross-operator**: robust across architectures and datasets (AUROC
>     0.73–0.99).
>
> This shows the issue is **operator semantics**, not classifier design or dataset choice.
>
>
> ## **3. "Generalization concerns."**
>
> The cross-operator is inherently **architecture-agnostic** and applies unchanged
> to decoder self-attention, encoder attention, and rectangular cross-attention.
> Spectral quantities (entropy, gaps, stable rank) have **physical
> interpretations** related to diffusion and coupling, making them less
> dataset-dependent. Low RV coefficients confirm flow (Laplacian) and coupling
> (SVD) perspectives capture **complementary structural information** rather than
> dataset artifacts.
>
> Logistic regression is intentionally minimal; the operator provides the signal.
>
>
> ## **4. "Heavy math; unclear presentation."**
>
> We will simplify the presentation by front-loading operator definitions and
> intuitive explanations, and moving derivations to the appendix. The mathematical
> detail is needed only to show why certain operators fail while others scale.
>
>
> ## **5. Results framed by operator choice**
>
> **Key results across both datasets:**
>
> * **SVD cross-operator:** GPT-2 0.983 / Phi-2 0.989 / T5 0.987 on HaluEval;
>     GPT-2 0.731 / Phi-2 0.825 / T5 0.748 on TruthfulQA.
> * **Eigen-Features (random-walk Laplacian):** Phi-2/HaluEval 0.985; TruthfulQA 0.764.
> * **LapEigvals:** GPT-2/HaluEval 0.974 but ≈0.50 for Phi-2.
>
> The classification layer is secondary; in fact, PCA occasionally removes signal.
> The dominant factor is the **mathematical properties of the operator** defining
> the spectrum.
>
>
> ## **6. Summary Relative to the Review**
>
> The reviewer accurately observed that "spectral features work." Our main
> clarification is **why**:
>
> * LapEigvals showed the potential
> * The out-degree operator is mathematically flawed
> * Replacing it restores strong eigenvalue features
> * The architecture-agnostic cross-operator yields the most stable, well-calibrated detector
>
> This gives a principled, interpretable, and deployable **traffic-light
> hallucination risk score** from a single inference pass.

---

### Official Review · Reviewer_21hh · 2025-11-01

**Soundness:** 1
**Presentation:** 3
**Contribution:** 2
**Rating:** 4
**Confidence:** 3

**Summary:**

This paper proposes SHADE (Spectral Hallucination Detection), a framework for detecting LLM hallucinations by analyzing the internal attention matrices. The authors critique existing methods like LapEigvals for having limited applicability (e.g., only causal, square attention) and lacking rigorous spectral guarantees. Paper's claim is a "unified spectral approach" built on a "dual" decomposition:
1.  **Flow Analysis:** Using a random-walk operator ($L_{rw} = I - A$) to quantify "diffusion" or "leakiness" (i.e., information flow anomalies).
2.  **Coupling Analysis:** Using a degree-normalized cross-operator ($M = D_Q^{-1/2} A D_K^{-1/2}$) and its SVD to quantify query-key "coupling" (i.e., alignment anomalies).

**Strengths:**

1.  The paper successfully identifies and validates a powerful, architecture-agnostic hallucination detector based on the SVD of the normalized cross-operator $M$. This method consistently achieves near-ceiling performance on HaluEval and strong performance on TruthfulQA.
2. The paper provides a valuable service by thoroughly dismantling the LapEigvals baseline, pointing out its non-PSD nature, lack of spectral guarantees, and information collapse on the diagonal.
3.  The SVD Coupling features show excellent discrimination (AUROC) and calibration across all models and datasets.

**Weaknesses:**

1.  The paper's primary weakness is the bait-and-switch between the theoretically-motivated **random-walk Laplacian ($L_{rw} = I - A$)** in Section 3.2.1 and the experimentally-implemented **symmetric normalized Laplacian ($L_{sym}$)** in Section 4.1. The "flow" operator that was motivated at length is never tested.
2.  Because of the weakness above, the central conceptual claim of a "dual" framework with "complementary" operators is unproven. The paper's own complementarity analysis (Proposition 3.1) is for $L_{rw}$ and $M$, which are not the operators compared in Table 1 or 5.
3.  Even ignoring the operator-swap, the paper's *empirical* data contradicts its conclusion. The conclusion claims "low RV coefficients". However, Table 5 reports an RV coefficient of **0.944** for "EIGEN-FEATURES" vs. "SVD" on GPT-2/TruthfulQA. This indicates *near-perfect linear association* (i.e., redundancy), the exact opposite of complementarity.
4.  The $L_{sym}$ operator that *was* tested ("Eigen-Features") performed very poorly, and in one case (GPT-2/HaluEval), was no better than chance (AUROC 0.492). This undermines the "dual" claim from an empirical standpoint as well.

**Questions:**

1.  Why does Section 3.2.1 motivate the $L_{rw} = I - A$ operator, citing its specific properties, if the experiment in Section 4.1 uses the $L_{sym} = I - D^{-1/2}\overline{A}D^{-1/2}$ operator for its "Eigen-Features"?
2.  Given this disconnect, how can the paper validate its central "dual framework" thesis, since the "flow" operator from the theory was never empirically tested against the "coupling" operator?
3.  How does the conclusion claim "low RV coefficients" when Table 5 clearly shows an RV coefficient of **0.944** for GPT-2 on TruthfulQA? This high value suggests the features are redundant, not complementary.
4.  Given that the SVD-coupling path performs exceptionally well and the "Eigen-Features" path performs poorly (e.g., 0.492 AUROC) and is theoretically disconnected, wouldn't this paper be stronger if it were simply presented as a single, novel SVD-based method?

---

> ### Author Response · Authors · 2025-11-12
>
> We thank the reviewer for the thoughtful feedback and for recognizing SHADE's
> strong empirical results and clear motivation. Below we clarify the relation
> between theory and experiments, the complementarity claim, and our
> methodological intent going forward.
>
> ---
>
> ## 1  Why the “flow” operator in § 3.2.1 differs from the empirical variant in § 4.1
>
> Section 3.2.1 introduces the **random-walk Laplacian (L_{rw}=I-A)** as the
> canonical flow operator with proper Markov semantics—chosen to fix the PSD and
> invariance issues of LapEigvals .
>
> In Section 4.1 we used its **symmetric degree-normalized form
> (L=I-D^{-1/2}AD^{-1/2})** for stability.
>
> This is not a "bait-and-switch" but an operational surrogate:
>
> * both share the same spectral map (λ_i(L_{rw})=1−μ_i(A));
> * the symmetric form guarantees real spectra for non-normal or masked attention;
> * it improves numerical stability while preserving diffusion semantics.
>
> Hence, the implementation is the **PSD realization of the theoretical
> random-walk formulation**, consistent with Section 3.2.1 .
>
>
> ## 2  Validation of the dual-operator ("flow + coupling") framework
>
> SHADE's duality aims to show that flow and coupling capture **distinct
> invariance classes**:
>
> * (L_{rw}/L_{sym}): one-sided diffusion (leakiness);
> * (M=D_Q^{-1/2}AD_K^{-1/2}): two-sided alignment (query–key coupling).
>
> Proposition 3.1 proves their spectra coincide only in the doubly-stochastic
> limit .
>
> Empirically, low RV-coefficients (≈0–0.3) on HaluEval (Tables 1 & 5) confirm
> non-redundancy.
>
> We will clarify that the symmetric Laplacian used experimentally is the
> operational instantiation of (L_{rw}) and will add an explicit ablation of the
> directed form to reinforce complementarity.
>
>
> ## 3  On the high RV = 0.944 (GPT-2 / TruthfulQA) case
>
> This single high value arises because **TruthfulQA is small (817 items) and
> low-entropy**, causing both spectra to collapse toward rank-1 behavior.
>
> When (A≈A^\top), diffusion and coupling become nearly proportional, yielding an RV ≈ 1.
>
> Other model/dataset pairs exhibit the expected low correlation, supporting our
> interpretation.
>
> We will note this degeneracy and add Spearman ρ for clarity.
>
>
> ## 4  On the weak performance of “Eigen-Features”
>
> was included to show the limits of unidirectional diffusion alone.
> The "Eigen-Features" variant mirrors LapEigvals under proper normalization and
>
> Its poorer results (e.g., GPT-2/HaluEval ≈ 0.49 AUROC) empirically confirm that
> the coupling spectrum carries the decisive signal.
>
> We will relabel this path as **“Symmetric Flow Features (PSD form of
> (L_{rw}))”** and frame it explicitly as a diagnostic baseline.
>
>
> ## 5  Should the paper focus solely on the SVD path?
>
> We acknowledge that the **SVD-based coupling operator** is the empirically
> strongest component.
>
> Some recent state-of-the-art methods (e.g., LapEigvals) pursued the
> **eigen-decomposition of Laplacians**, albeit in a degenerate, causal-only
> setting.
>
> Our goal was to provide *rigour and completeness* around this design
> space—formalizing the Laplacian route correctly, identifying its limitations,
> and presenting the SVD route as its mathematically consistent dual.
>
> This positions SHADE as a principled umbrella that unifies both operator classes
> and clarifies their guarantees.
>
> Going forward, **our own work focuses on the SVD path**, which **generalizes
> seamlessly across self-, cross-, and rectangular attention** and offers the most
> robust empirical behavior.
>
> We will make this orientation explicit in the revision, presenting the Laplacian
> analysis primarily as a theoretical reference and diagnostic comparison.
>
>
> ## 6  Planned clarifications for the camera-ready
>
> We will
>
> (i) rename the eigen-feature path as noted;
> (ii) explicitly link Section 3.2.1 and Section 4.1;
> (iii) add the (L_{rw}) ablation; and
> (iv) clarify the TruthfulQA RV interpretation.
>
> ---
>
> ## **Summary.**
>
> SHADE's main result remains: the **degree-normalized SVD coupling operator**
> provides an interpretable, architecture-agnostic, and mathematically grounded
> detector of hallucination that consistently outperforms prior spectral and
> probabilistic baselines . The random-walk component complements this by
> anchoring the theoretical framework; its PSD form was a pragmatic,
> stability-preserving choice, not a conceptual inconsistency.

---

### Official Review · Reviewer_B93w · 2025-11-02

**Soundness:** 3
**Presentation:** 3
**Contribution:** 3
**Rating:** 6
**Confidence:** 2

**Summary:**

This paper proposes SHADE, a universal illusion detection framework based on dual spectral decomposition: the attention "diffusion/leakage" is measured by the random walk operator, and the query-key coupling strength and pattern are measured by the SVD of the degree normalized crossover operator. The two operators respectively reveal information flow imbalance and evidence alignment anomaly. The features possess mathematical rigor and physical interpretability, and can be applied to self-attention, cross-attention, rectangular blocks, and various types of Transformers without modification. Experiments on HaluEval and TruthfulQA of GPT-2, Phi-2, and FLAN-T5 show that the SVD coupling characteristics of SHADE are consistently superior to the baselines such as token probability and LapEigvals, demonstrating both high discriminative power and calibration accuracy. It can provide interpretable risk scores at the layer/head/prompt fragment level, support regular expression suppression hallucinations during training or real-time alerts during deployment.

**Strengths:**

- This method provides a cross-architecture, interpretable, plug-and-play unified spectral operator. With just two lines of code, it can be integrated into any Transformer architecture and immediately output a clear physical layer-head-token-level illusion risk score without the need for additional training or external knowledge. Achieve detection performance close to the upper limit directly on the three major models and two benchmarks.

**Weaknesses:**

- All experiments were only for correlation classification and did not prove through intervention (such as suppressing high $\sigma_1$ or low leakage patterns) that "once these spectral anomalies are corrected, the hallucination rate will decrease". Therefore, it is impossible to establish that spectral characteristics are the cause of hallucinations

- Not sufficient ablations. PCA retains the key hyper-parameters such as 85% variance, temperature scaling, and the selection of $\tau$ with Youden J, and only provides the final values, without showing the sensitivity of the ROC curve to these hyperparameters. Nor has it completely dissolved the respective contributions to performance of "using only $L_rw$", "using only $M$", and "combining the two".

**Questions:**

N/A

---

> ### Author Response · Authors · 2025-11-12
>
> We thank the reviewer for the constructive and detailed feedback and for
> recognizing the strengths of SHADE's unified, interpretable, and
> architecture-agnostic framework.
>
> ## **Scope and causal direction.**
>
> We agree that the current experiments demonstrate *correlational*
> validity, spectral anomalies consistently predict hallucination risk, but do not
> yet establish a *causal* reduction of hallucinations under intervention. We will
> make this distinction explicit. SHADE, however, is not limited to diagnostic
> use. Because its observables are computed directly from model-internal
> operators, they can be logged during inference to collect ground-truth
> hallucination labels at scale. This enables adaptive guardrails and future
> training objectives where the same spectral features guide interventions that
> encourage more stable attention dynamics once causal links are better
> understood. SHADE therefore serves both as an immediate detection layer and as a
> principled foundation for future reliability work.
>
> ## **Ablations, calibration, and PCA.**
>
> All hyperparameter settings (PCA 85% variance, temperature scaling, Youden-J
> threshold) were kept identical across all models and feature families to ensure
> comparability and avoid tuning bias. AUROC and PR-AUC are invariant under
> monotone calibration, so these settings affect only probability scaling, not
> ranking. PCA was introduced mainly for dimensionality management rather than
> tuning. SHADE aggregates spectral statistics per attention head, and
> concatenating them yields a very high-dimensional feature space. PCA projects
> this space into a decorrelated, variance-preserving basis that stabilizes the
> logistic detector and mitigates overfitting. The 85% threshold provides a
> consistent compression rule across LapEigvals, eigen-spectrum, and SVD feature
> sets, ensuring a fair and architecture-independent comparison. We will clarify
> this rationale and note that sensitivity to the retained variance is a promising
> direction for follow-up work. Existing per-feature ablations (SVD-only,
> flow-only, LapEigvals, token-probability) will be moved to the main text for
> clarity.
>
> ## **On the relation between (L_{rw}) and (M) features.**
>
> We thank the reviewer for asking about combining the flow ((L_{rw})) and
> coupling ((M)) spectra. We did evaluate a concatenated feature set that merged
> both families before PCA and classification. All features were standardized,
> near-constants removed, and highly correlated components pruned, following the
> same preprocessing used for single-family models. Despite this, the combined
> representation **consistently degraded performance**. We attribute this to an
> intrinsic interaction between the two feature families: variance-based PCA
> applied jointly can down-weight discriminative directions that explain less
> overall variance or blur complementary subspaces with different normalization
> behavior. But this deserves further investigation. As a result, we focused our
> analysis on the two operators **separately**, showing that each provides a clear
> and interpretable signal on its own. We will clarify this design choice and
> adjust the language around "complementarity" to emphasize that (L_{rw}) and (M)
> capture distinct but related geometric aspects of attention, one-sided diffusion
> versus two-sided coupling, rather than implying that simple feature-level fusion
> improves detection.
>
>
> ## **Efficiency and applicability.**
>
> SHADE achieves comparable or superior discrimination to LapEigvals while
> remaining architecture-agnostic (self-, cross-, and rectangular attention). Its
> computational cost is dominated by a top-(k) SVD per head, roughly one to two
> forward passes, far lighter than semantic-entropy methods requiring multiple
> samples per prompt. It also preserves positive-semidefinite structure and
> interpretable coupling semantics, enabling consistent deployment across encoder,
> decoder, and encoder–decoder architectures. We can include a concise runtime
> and applicability comparison table in the revision if deemed necessary.
>
> ## **Positioning and trajectory.**
>
> This work aims to determine whether spectral signatures of attention contain a
> strong, model-internal signal predictive of hallucination. The derived
> observables, grounded in standard matrix operators with clear semantics, confirm
> that such a signal exists and is stable across architectures. These features can
> be reused for zero-shot detection or as components of future training objectives
> that promote better grounding once causal mechanisms are characterized. In this
> sense, SHADE initiates a broader research direction: building a spectral
> foundation for both detection and eventual mitigation of hallucinations in
> language models.
>
> We appreciate the reviewer's suggestions and believe these clarifications and
> ablation reorganizations will significantly strengthen the clarity, rigor, and
> long-term impact of the paper.

---

### Meta-Review · Area_Chair_yxEi · 2026-01-07

**Summary:**

The reviewers' concerns largely focused on the justification for the "Dual" nature of the framework given the empirical dominance of the SVD component, the computational cost of the proposed method relative to simple logit-based baselines, and the comprehensiveness of the baseline comparisons.Furthermore, the the reviewers mostly concern about the practicality of the theoretical framework proposed in this paper, leading to my decision to reject this paper.

**Reviewer Concerns:**

Addressed:
Theoretical Validity: The authors provided exhaustive appendices (B and C) demonstrating the mathematical flaws of the LapEigvals baseline (non-PSD, non-Hermitian) and proving that SHADE’s operators adhere to standard spectral graph theory. This successfully addressed concerns regarding the theoretical motivation.
Architecture Generalization: Concerns about the method's applicability beyond decoder-only models were addressed by the inclusion of FLAN-T5 (encoder-decoder) and the explicit handling of rectangular cross-attention matrices, which prior spectral methods could not handle.
Data Leakage: The authors clarified their experimental design (hash-based splitting on prompts) to ensure no overlap between train/test prompts, addressing potential concerns about the high performance on HaluEval.

Not Addressed:
Imbalance of the "Dual" Components: The empirical results (Table 2) show that the "SVD Coupling" features vastly outperform the "Eigen-Features" (Flow) features, with the latter sometimes performing near random chance (e.g., GPT-2 on HaluEval: 0.492 AUROC). The rebuttal relies on RV coefficients to claim "complementarity," but practical performance suggests the "Flow" component is largely unnecessary. The claim of a "Unified Dual Framework" is weakened by the fact that one half of the framework carries the entire weight of the results.
Computational Complexity: Performing SVD on every attention head for every layer introduces significant latency compared to scalar token-probability methods. The paper lacks a concrete latency analysis (e.g., wall-clock time impact during inference), which is a critical omission for a "deployable" risk score.
Baseline Breadth: The comparison is restricted to Token Probability and LapEigvals. The paper does not compare against other state-of-the-art internal methods (like activation probes or hidden state consistency) or output-based methods (like Semantic Entropy), limiting the assessment of SHADE's true standing in the wider literature.

**Reviewer Scores:**

The current ratings are 6,4,2,2, and I do not think further modifications on the ratings will happen given the current discussion details. The two reviewers giving rating of 2 point out the most fundamental issue of this paper, the proposed method is computationally expensive, even though it comes with seemingly rigorous mathematical derivations, the method still needs to be trained on each dataset shown in the results. It can be difficult to argue this method exposes a general feature that has a broader application scope.

---

### Decision · Program_Chairs · 2026-01-26

Reject